# Courses of depressive symptoms and diabetes incidence among middle-aged and older adults: A prospective study

**Rachel J. Burns** [ID]*, **Katherine Ford, Geneviève C. Forget, Kimia Fardfini-Ruginets, Richard Ward**

Department of Psychology, Carleton University, Ottawa, Canada

* rachel.burns@carleton.ca

## Abstract

Elevated depressive symptoms are a risk factor for diabetes. Although depressive symptoms can remit or emerge over time, little work has considered if courses of depressive symptoms are associated with incident diabetes. The purpose of this study was to explore associations between courses of depressive symptoms and incident diabetes. Data came from the English Longitudinal Study of Ageing ($n = 4,978$), which is an ongoing, cohort study of adults aged 50 years and older residing in private households in England. Depressive symptoms were measured biennially from 2002 to 2008. Participants were categorized into one of six groups: no depressive symptoms, remitted depressive symptoms, incident depressive symptoms with remission, incident depressive symptoms without remission, chronic depressive symptoms, and variable course. Diabetes status was self-reported biennially from 2010 to 2018. After adjusting for covariates, remitted depressive symptoms ($HR = 1.52$, $95\%$ $CI$ [1.06, 2.22]) and variable course depressive symptoms ($HR = 1.83$, $95\%$ $CI$ [1.19, 2.81]) remained associated with incident diabetes. In sensitivity analyses, which lowered the cut-off score for depressive symptoms, variable course depressive symptoms ($HR = 1.61$, $95\%$ $CI$ [1.11, 2.33]) remained associated with incident diabetes. Specific courses of depressive symptoms, including variable course depressive symptoms, were associated with diabetes incidence. Continuing to examine the link between patterns of depressive symptoms over time and incident diabetes may lead to the development of more targeted interventions.

## Introduction

Approximately 10% of adults worldwide have diabetes [1]. Diabetes is more common among middle-aged and older adults relative to younger age groups. For example, in the United Kingdom, where the prevalence of diabetes among adults is approximately 10%, 5% of adults under age 45 years, 9% of adults aged 45–64 years, and 21% of adults aged 65 years and older have diabetes [2]. Depressive symptoms are a risk factor for diabetes; according to the most recent meta-analysis, depression —as measured at one timepoint—is associated with an 18% increased risk of incident type 2 or adult-onset diabetes [3]. However, the vast majority of

**Data availability statement:** The data used in this study are from the English Longitudinal Study of Ageing (ELSA), which is an ongoing cohort study of approximately 20,000 older adults. Data are publicly available, without fees, on the UK Data Service website to users who agree with their conditions on use, which, at the time of data access, included: 1. I agree not to link nor attempt to link the ELSA Wave 0 data to the Health Survey for England (HSE) data 2. I agree not to use the Wave 0 data in any way to identify participants from ELSA or HSE 3. I agree not to use nor attempt to use ELSA data to identify specific geography from which the study sample was selected, nor to claim to have done so Data available at: https://beta.ukdataservice.ac.uk/datacatalogue/studies/study?id=5050

**Funding:** RB holds a Canadian Institutes of Health Research (CIHR) Project grant (PJT-173303) that partially supported this work. The funder played no role in study design, data collection/analysis, decision to publish, or preparation of the manuscript. https://www.cihr-irsc.gc.ca/. KF-R is supported in part by funding from the Social Sciences and Humanities Research Council (752-2023-2541). The funder played no role in study design, data collection/analysis, decision to publish, or preparation of the manuscript. https://www.sshrc-crsh.gc.ca/.

**Competing interests:** The authors have declared that no competing interests exist.

studies in this area have measured depressive symptoms at one time point; few studies consider if the course of depressive symptoms over time is linked to diabetes incidence.

Depressive symptoms may remit or emerge and are thus not always stable over time. For example, in a community sample of middle-aged and older adults, Beekman et al.[4] identified six courses of depression: never depressed, incident depression with remission, incident depression without remission, remitted depression, chronic depression, and variable course depression. Patterns of depressive symptoms over time have been linked to several physical health outcomes. For example, trajectories of depressive symptoms differentially predict incident dementia [5,6] and mortality [7]. In particular, the persistence, remittance, and onset of depressive symptoms have been differentially linked to physical health outcomes. Among older adults, chronic depressive symptoms and the onset of depressive symptoms were associated with declines in physical performance, but remitted depressive symptoms were not [8]. Chronic depressive symptoms, new-onset depressive symptoms, and remitted depressive symptoms were associated with excess mortality, albeit with differing levels of risk, and chronic depressive symptoms and remitted depressive symptoms were associated with incident cardiovascular disease [9]. Remitted depressive symptoms are also associated with improved functional ability [10].

Behavioural (e.g., physical activity, diet) and physiological (e.g., HPA axis dysregulation, increased inflammatory responses, weight gain which may or may not be related to antidepressant use) mechanisms are posited to mediate the link between depression and incident diabetes [11,12]. Specific courses of depressive symptoms (e.g., chronic depressive symptoms, variable course) may possibly result in longer term exposure to these mechanisms than other courses (e.g., incident depressive symptoms with remission) thereby differentially affecting diabetes risk. Indeed, limited work suggests that patterns of depressive symptoms over time are linked to diabetes incidence. Carnethon et al. [13] concluded that elevated depressive symptoms at one timepoint, elevated depressive symptoms at two timepoints, and an increase in depressive symptoms were associated with diabetes incidence relative to never having depressive symptoms. Similarly, two studies have demonstrated that having depression at both baseline and follow-up was associated with an increased risk of diabetes [14,15]. However, these studies focused on chronic depressive symptoms as measured at two timepoints and thus did not study courses of depressive symptoms characterized by the remittance or onset of symptoms.

A more recent study in this area used a data-driven approach (i.e., group-based trajectory modeling) to identify five trajectories of depressive symptoms characterized by no depressive symptoms, low depressive symptoms, low-moderate depressive symptoms, moderate depressive symptoms, and elevated yet increasing depressive symptoms. Relative to the no depressive symptoms trajectory group, all other groups were at higher risk of incident diabetes. Moreover, the risk of diabetes seemed graded such that trajectories characterized by greater depressive symptoms were associated with greater risk of diabetes incidence [16]. This study demonstrated the value of considering longitudinal patterns of depressive symptoms as diabetes risk factors, but its data-driven approach to identifying trajectories of depressive symptoms had limitations. Namely, the data-driven analytic approach yielded trajectories of depressive symptoms that were mostly stable over time, which is typical when using data-driven approaches to identify trajectories of depressive symptoms [17]. As a result, it is unknown if courses of depressive symptoms that involve the remittance or onset of depressive symptoms are related to incident diabetes.

Therefore, the purpose of this study was to explore if courses of depressive symptoms characterized by the remittance and onset of depressive symptoms were associated with incident diabetes. Specifically, associations between courses of depressive symptoms identified

by Beekman et al. [4] (i.e., no depressive symptoms, incident depressive symptoms with remission, incident depressive symptoms without remission, remitted depressive symptoms, chronic depressive symptoms, and variable course symptoms) and incident self-reported diabetes during a 10 year follow-up were examined in middle-aged and older adults.

## Materials and methods

### Data Source

Data came from the English Longitudinal Study of Ageing (ELSA), which is an ongoing, cohort study that includes adults aged 50 years and older and their cohabiting partners residing in private households in England. ELSA is funded by the National Institute on Aging (R01AG017644), and by UK Government Departments coordinated by the National Institute for Health and Care Research (NIHR). The ELSA cohort was generally representative of the English population at baseline and is further detailed by Steptoe et al. [18]. Baseline data were collected in 2002/03 and data collection is ongoing. Data on a range of topics, including mental and physical health, are collected every two years via face-to-face interviews, paper-based questionnaires, and nurse visits. The National Health Service (NHS) Research Ethics Committees approved each wave of the ELSA protocol. Carleton University Research Ethics Board-B also approved these analyses before they were initiated (Project # 117637). All participants provided informed consent verbally to participate in the interview portion of ELSA and in writing for other aspects of the study, such as blood draws and administrative linkages. Verbal consent was witnessed and documented by trained interviewers and fieldworkers. Data are publicly available through the UK Data Service [19].

In the present study, the exposure period, during which depressive symptoms were measured, spanned waves 1–4 of ELSA (i.e., 2002/03, 2004/05, 2006/07, 2008/09). The follow-up period, during which diabetes status was measured, spanned waves 5–9 (i.e., 2010/11, 2012/13, 2014/15, 2016/17, 2018/19). Depressive symptoms and diabetes status were assessed during face-to-face interviews. The current study included only participants who (a) self-reported not having diabetes in wave 4 ($n = 10,033$), (b) provided data on diabetes status at least once during the follow-up period ($n = 8,920$) and (c), provided complete data on measures of depressive symptoms during the waves 1–4 (i.e., the exposure period), resulting in a sample of 4,978 participants. Included participants tended to be younger and were more likely to be white, married/partnered, have less than college degree, and be women (all $p<.05$) compared to ELSA participants who self-reported not having diabetes in wave 4, but were excluded.

### Measures

**Courses of depressive symptoms.**  Depressive symptoms were measured at waves 1–4 with a modified, 8-item version of the Centre for Epidemiologic Studies-Depression Scale (CES-D) [20]. Items assessed symptoms of depression (e.g., felt depressed, restless sleep, felt everything was an effort) experienced in the past week (response options: *yes*, *no*). Two of the eight items (i.e., was happy, enjoy life) were reversed scored. Affirmative responses were summed to create a total depressive symptom score at each time point ($KR\text{-}20_{waves1–4} = .78\text{-}.79$).

Total scores were then used to identify each participant's depressive symptoms status (i.e., no depressive symptoms, depressive symptoms) at each time point. In the main analyses, individuals with total scores of 4 or more were classified as having depressive symptoms and those with scores of 3 or less were classified as not having depressive symptoms based on previous validation work [20]. However, scores of 3 or more are also often used as an indication of depressive symptoms with this measure [20–22]. Thus, sensitivity analyses were conducted in

which individuals with scores of 3 or more were classified as having depressive symptoms and those with scores of 2 or less were classified as not having depressive symptoms.

Based on their depressive symptoms status at each time point (i.e., no depressive symptoms, depressive symptoms), participants were then categorized into one of six patterns of depressive symptoms specified by Beekman et al. [4]: (1) *no depressive symptoms*, defined as not having depressive symptoms at waves 1–4; (2) *incident depressive symptoms with remission*, defined as not having depressive symptoms at wave 1, followed by at least one wave of depressive symptoms, and not having depressive symptoms at wave 4; (3) *incident depressive symptoms without remission*, defined as not having depressive symptoms at wave 1, followed by at least one wave of depressive symptoms that do not remit by wave 4; (4) *remitted depressive symptoms*, defined as having depressive symptoms at wave 1 that remit by wave 4; (5) *chronically elevated depressive symptoms*, defined as having depressive symptoms at all waves; and (6) *variable course*, defined by too many transitions in depressive symptoms status to be otherwise classified.

**Diabetes status.** Self-reported diabetes status was measured at waves 4–9. Participants were asked if they had ever been told by a doctor that they have diabetes or high blood sugar (*yes*, *no*).

**Covariates.** Potential confounders, which were suspected to affect both courses of depressive symptoms and incident diabetes, were selected as covariates [23]. All covariates were measured at baseline and included self-reported age, sex, education (less than college degree; college degree), marital status (married/partnered; single/widowed/divorced), and ethnicity (white; not white). Potential mediators, which may lie on the causal pathway between courses of depressive symptoms and incident diabetes, were not included as covariates to avoid overcontrolling [23]. For example, health behaviours (e.g., physical activity) and some physiological processes (e.g., weight gain, blood pressure) are posited to mediate the association between depression and incident diabetes, are were thus not included as covariates [11,24].

## Statistical analyses

For descriptive purposes, sex, ethnicity, marital status, education, age, and the percentage of participant who developed diabetes during follow-up were compared across courses of depressive symptoms via chi-square tests and one-way ANOVAs for categorical and continuous variables, respectively. Cox proportional hazard models tested the association between courses of depressive symptoms and incident diabetes. Clustered robust standard errors were used because 52.69% of participants shared households at baseline. Participants lost to follow-up or who were alive at the end of the follow-up period were censored. Courses of depressive symptoms were dummy coded; the no depressive symptoms course was the referent. An unadjusted model and a model adjusting for covariates (i.e., sex, ethnicity, marital status, education, age) were run. One hundred and one (2.03%) participants were missing covariate data and were excluded from the adjusted model. Losing a small number of cases (i.e., less than approximately 5%) due to missing data is unlikely to result in bias [25]. A sensitivity analysis was conducted in which analyses were repeated with courses of depressive symptoms based on CESD scores of 3 or more indicating depressive symptoms [20–22]. Data preparation was conducted with SPSS for Windows version 28 [26] and analyses were conducted with Stata 16 [27].

## Results

Of the 4,978 participants, 71.13% ($n = 3,541$) belonged to the no depressive symptoms group, 6.47% ($n = 322$) belonged to the remitted depressive symptoms group, 8.60% ($n = 428$)

belonged to the incident depressive symptoms with remission group, 6.45% ($n = 321$) belonged to the incident depressive symptoms without remission group, 4.70% ($n = 234$) belonged to the variable course group, and 2.65% ($n = 132$) belonged to the chronic depressive symptoms group in the main analysis, in which having depressive symptoms was defined as a score of 4 or higher on the CES-D. Participant characteristics by depressive symptoms course are presented in Table 1.

During follow-up, 6.99% ($n = 348$) of participants reported developing diabetes ($M_{time\ at\ risk} = 3.99$ waves). Table 1 shows the percentage of participants in each depressive symptoms group who reported developing diabetes. In the main analysis, in which depressive symptoms were defined as a score of 4 or higher on the CESD, the unadjusted model indicated that remitted depressive symptoms, $HR = 1.50$, 95% CI [1.03, 2.17], variable course depressive symptoms, $HR = 1.89$, 95% CI [1.26, 2.85], and chronic depressive symptoms, $HR = 1.82$, 95% CI [1.06, 3.14], were associated with incident diabetes (Table 2). Incident depressive symptoms with remission and without remission were not associated with incident diabetes (Table 2). Although remitted depressive symptoms, $HR = 1.53$, 95% CI [1.06, 2.22], and variable course depressive symptoms, $HR = 1.83$, 95% CI [1.19, 2.81], remained associated with incident diabetes after adjusting for covariates, chronic depressive symptoms did not, $HR = 1.67$, 95% CI [.95, 2.94] (Table 2).

In the sensitivity analysis, in which having depressive symptoms was defined as a score of 3 or higher on the CESD, 60.00% of participants ($n = 2,987$) belonged to the no depressive symptoms group, 8.74% ($n = 435$) belonged to the remitted depressive symptoms group, 10.57% ($n = 526$) belonged to the incident depressive symptoms with remission group, 8.50% ($n = 423$) belonged to the incident depressive symptoms without remission group, 7.31% ($n = 364$) belonged to the variable course group, and 4.88% ($n = 243$) belonged to the chronic depressive symptoms group. The unadjusted and adjusted models of the sensitivity analysis indicated that variable course depressive symptoms, $HR_{adjusted} = 1.61$, 95% CI [1.11, 2.33], and chronic depressive symptoms, $HR_{adjusted} = 1.77$, 95% CI [1.15, 2.72], were associated with incident diabetes (Table 2).

**Table 1. Participant characteristics by course of depressive symptoms group (main analyses; CESD ≥ 4).**

| | Total sample ($n = 4,978$) | Never depressive symptoms ($n = 3,541$) | Remitted depressive symptoms ($n = 322$) | Incident depressive symptoms with remission ($n = 428$) | Incident depressive symptoms without remission ($n = 321$) | Variable course depressive symptoms ($n = 234$) | Chronic depressive symptoms ($n = 132$) | p value of chi square/ F test comparing groups |
|---|---|---|---|---|---|---|---|---|
| % Women ($n$) | 58.86 (2,930) | 53.74 (1,903) | 68.94 (222) | 67.76 (290) | 72.27 (232) | 76.92 (180) | 78.03 (103) | <.001 |
| % White ($n$) | 98.45 (4,901) | 99.10 (3,509) | 95.96 (309) | 97.43 (417) | 97.82 (314) | 96.58 (226) | 95.45 (126) | <.001 |
| % Married/ living with partner ($n$) | 73.56 (3,661) | 77.94 (2,759) | 62.11 (200) | 67.06 (287) | 68.22 (219) | 55.98 (131) | 49.24 (65) | <.001 |
| % College degree, ($n$) | 22.89 (1,116) | 25.42 (883) | 17.89 (56) | 17.97 (76) | 16.35 (51) | 17.62 (40) | 7.75 (10) | <.001 |
| Age (years), $M$ ($SD$) | 61.52 (9.31) | 61.30 (9.12) | 61.11 (9.16) | 61.53 (9.49) | 63.42 (10.12) | 62.47 (9.79) | 62.03 (10.67) | .002 |
| % Developed diabetes during follow-up ($n$) | 6.99 (348) | 6.33 (224) | 9.32 (30) | 7.94 (34) | 6.85 (22) | 10.68 (25) | 9.85 (13) | .031 |

*Notes:* Characteristics assessed at baseline unless otherwise stated. *p*-values were calculated for chi-squaretests for categorical variables and one-way ANOVA for continuous variables.

**Table 2. Associations between courses of depressive symptoms and incident diabetes.**

| | Hazard Ratio [95% CI] | |
|---|---|---|
| **Main Analysis (CESD ≥ 4)** | | |
| | **Unadjusted Model** | **Adjusted Model** |
| Never depressive symptoms (*n* = 3541) | referent | referent |
| Remitted depressive symptoms (*n* = 322) | 1.50 [1.03, 2.17] | 1.53 [1.06, 2.22] |
| Incident depressive symptoms with remission (*n* = 428) | 1.33 [.92, 1.93] | 1.34 [.93, 1.94] |
| Incident depressive symptoms without remission (*n* = 321) | 1.23 [.79, 1.90] | 1.30 [.84, 2.02] |
| Variable course depressive symptoms (*n* = 234) | 1.89 [1.26, 2.85] | 1.83 [1.19, 2.81] |
| Chronic depressive symptoms (*n* = 132) | 1.82 [1.06, 3.14] | 1.67 [.95, 2.94] |
| **Sensitivity Analysis (CESD ≥ 3)** | | |
| | Unadjusted Model | Adjusted Model |
| Never depressive symptoms (*n* = 2,987) | referent | referent |
| Remitted depressive symptoms (*n* = 435) | 1.27 [.89, 1.82] | 1.31 [.91, 1.87] |
| Incident depressive symptoms with remission (*n* = 526) | 1.03 [.71, 1.50] | 1.05 [.72, 1.51] |
| Incident depressive symptoms without remission (*n* = 423) | 1.38 [.96, 1.99] | 1.45 [1.01, 2.08] |
| Variable course depressive symptoms (*n* = 364) | 1.61 [1.13, 2.30] | 1.61 [1.11, 2.33] |
| Chronic depressive symptoms (*n* = 243) | 1.86 [1.22, 2.83] | 1.77 [1.15, 2.72] |

*Notes:* Adjusted model controlled for sex, ethnicity, marital status, education, and age.

## Discussion

This study was among the first to determine if courses of depressive symptoms over time were associated with incident diabetes thereby extending the literature, which has largely focused on depressive symptoms as measured at a single timepoint as a risk factor for incident diabetes. Courses of depressive symptoms were differentially associated with incident diabetes. In all analyses, variable course depressive symptoms were associated with incident diabetes. Although remitted depressive symptoms and chronic depressive symptoms were associated with incident diabetes in some analyses, this association was sensitive to the inclusion of covariates or the CES-D cut-off that was used to define elevated depressive symptoms. Courses characterized by incident depressive symptoms with or without remission were generally not associated with incident diabetes.

Depression may be an acute condition for some individuals, yet a chronic condition characterized by persistence or recurrence of symptoms for others [28]. However, depressive symptoms have generally not been conceptualized as such in prior work that examined associations between depressive symptoms and incident diabetes. Rather, work in this area has generally measured depressive symptoms at one timepoint [3,29,30]. Yet, depending upon when measurements are taken, elevated depressive symptoms detected at a single timepoint could be indicative of several courses of depressive symptoms, which may or may not be associated with incident diabetes. For example, an individual identified as having elevated depressive symptoms at a single measurement point may subsequently experience remission, chronic depressive symptoms, or alternating periods of depression and remission. Indeed, the present results demonstrated that some of the courses of depressive symptoms which contained at least one instance of elevated depressive symptoms (e.g., incident depressive symptoms with remission) were not associated with incident diabetes whereas other courses which also contained at least one instance of elevated depressive symptoms (e.g., variable course) were associated with incident diabetes, thereby demonstrating the value in considering patterns of depressive symptoms over time.

The general finding that courses of depressive symptoms were differentially associated with incident diabetes is consistent with work demonstrating associations between specific trajectories of depressive symptoms and other physical health outcomes, such as dementia and mortality [5,7]. Previous work in the context of diabetes, which used a data-driven approach to identify generally stable trajectories of depressive symptoms, demonstrated that four trajectories of depressive symptoms (i.e., low depressive symptoms, low-moderate depressive symptoms, moderate depressive symptoms, elevated and increasing depressive symptoms) were associated with incident diabetes relative to the no depressive symptoms group and that the group with elevated and increasing depressive symptoms was at highest risk of diabetes [16]. The latter finding is consistent with the present finding from the sensitivity analysis that chronically elevated depressive symptoms were associated with incident diabetes.

The data-driven approach to trajectory identification used in previous work yielded generally stable trajectories of depressive symptoms [16], which is a common issue in data-driven approaches. A review of studies using data-driven approaches to identify trajectories of depressive symptoms concluded that stable trajectories were more common than those characterized by change in depressive symptoms [17]. Therefore, data-driven approaches may not be optimal when seeking to examine the remittance or onset of depressive symptoms. By using predetermined courses of depressive symptoms characterized by remittance and onset, the current study was able examine courses that likely would not have emerged via data-driven approaches, such as variable course. As a result, the current study gleaned the novel findings that variable course depressive symptoms were associated with incident diabetes, but courses characterized by incident depressive symptoms with and without remission, generally, were not.

Comparison of the main and sensitivity analyses yielded several noteworthy findings. First, the association between variable course depressive symptoms and incident diabetes was robust across CES-D cutoffs. Second, associations between some courses of depressive symptoms and incident diabetes were detected even when a lower cut-off for the CES-D was used. Therefore, results suggest that some courses of relatively more moderate depressive symptoms are associated with incident diabetes. Third, some results differed when different CES-D cut-offs were used. In particular, remitted depressive symptoms were not associated with incident diabetes in sensitivity analyses in which a lower CES-D cut-off was used. Moreover, chronic depressive symptoms were associated with incident diabetes in both unadjusted models, but this association only held for the lower CES-D cut-off after controlling for covariates. Although this discrepancy could indeed accurately represent the underlying association, the sample size of the chronic depressive symptoms group in the main analyses ($n$ =132) was roughly half of that for the same group in the sensitivity analyses ($n$ = 243). The higher power that comes with larger sample sizes should increase confidence in the precision of effect estimates [31]. Indeed, the effect size estimates are comparable in the main and sensitivity analyses, though confidence intervals are much narrower for the chronic depression group in the sensitivity analyses. The lack of association between the chronic depressive symptoms group and incident diabetes in the main adjusted analysis, thus may be attributable to a lack of power. It is worth noting that we performed sensitivity analyses because a single cut-off for the version of the CES-D used in this study is not universally adopted, perhaps because as cut-off scores for the CES-D increase, specificity increases, but sensitivity declines [32].

Prior evidence suggests that the well-documented association between depressive symptoms and incident diabetes may be driven, at least in part, by behavioral and physiological mechanisms, including physical activity, diet, weight gain, hypothalamic-pituitary-adrenal (HPA) axis dysregulation, chronic activation of the sympathetic nervous system, and inflammation [11,12,24]. For example, health behaviours that are risk factors for diabetes, such as

low physical activity and poor diet, are often linked to depression and depression is linked to HPA axis dysregulation and chronic activation of the sympathetic nervous system, which increase blood pressure and inflammation thereby increasing T2D risk [11,12,24]. In variable course depressive symptoms, which was consistently associated with incident diabetes in the present study, depressive symptoms are experienced quite regularly. Therefore, results point to the possibility that *sustained* exposure to underlying physiological and/or behavioral mechanisms may further drive the association between depressive symptoms and incident diabetes. Investigating the mechanisms underlying this association is a promising avenue for future research.

This study has several strengths, including the use of a large, community-based sample, relatively long exposure and follow-up periods, and its novel focus on courses of depressive symptoms. However, this study also has several limitations. First, although the CES-D is widely used in population-based studies, it assesses depressive symptoms experienced over the past week. Participants may have experienced periods of elevated depressive symptoms, or lack thereof, between assessments. Relatedly, the CES-D is not a diagnostic interview, so results may not extend to courses of major depressive disorder. Similarly, in focusing on the experience of depressive symptoms, the use of pharmacological or psychological treatment for depression was not accounted for. Due to the predefined structure of the courses of depressive symptoms, participants were required to complete all four assessments of depressive symptoms to be included in the study. This may have introduced bias if individuals are less likely to participate in research studies when experiencing depression. ELSA does not measure diabetes type (e.g., type 1 or type 2). Given the age of participants in our sample, it is likely that the majority of incident diabetes cases were type 2 diabetes, however, we cannot rule out the possibility of some incident type 1 diabetes cases [33]. Diabetes status was assessed via self-reported physician diagnosis. Although good agreement between this measure and fasting plasma glucose has been demonstrated (81.5%) [34], results may not extend to those with undiagnosed diabetes. Finally, although ELSA participants are generally representative of the English population [18], the sample used in the present study was predominantly white, though this is reflective of the decision to not oversample racial or ethnic minority populations in ELSA due to financial constraints [18]. These limitations suggest clear directions for future work. For example, future studies examining the associations between courses of depressive symptoms and incident diabetes could include alternate assessments of depressive symptoms, such as diagnostic interviews and medical records, alternate measures of diabetes status, such as medical records and blood assays, and more diverse samples.

In conclusion, the results of this study suggest that variable course depressive symptoms were consistently associated with diabetes incidence whereas some other courses of depressive symptoms were not. Though work in this area has largely examined links between depressive symptoms measured at one timepoint and incident diabetes [29,30], this study suggests that the association between depressive symptoms and diabetes may be more nuanced. Continuing to examine the link between patterns of depressive symptoms over time and incident diabetes may lead to the development of more targeted diabetes prevention interventions. For example, if replication studies converge on similar findings, results may suggest that monitoring depressive symptoms over time in clinical settings may help to identify individuals to whom diabetes prevention programs are especially beneficial.

## Author contributions

**Conceptualization:** Rachel J Burns.

**Data curation:** Rachel J Burns, Katherine Ford, Kimia Fardfini-Ruginets, Richard Ward.

**Formal analysis:** Rachel J Burns.

**Investigation:** Rachel J Burns.

**Methodology:** Rachel J Burns.

**Supervision:** Rachel J Burns.

**Writing – original draft:** Rachel J Burns, Geneviève C. Forget.

**Writing – review & editing:** Rachel J Burns, Katherine Ford, Geneviève C. Forget, Kimia Fardfini-Ruginets, Richard Ward.

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
