## [Decision Letter · Decision Letter 0]

8 Dec 2024

PONE-D-24-44740Courses of depressive symptoms and diabetes incidence among middle-aged and older adultsPLOS ONE

Dear Dr. Burns,

Thank you for submitting your manuscript to PLOS ONE. After careful consideration, we feel that it has merit but does not fully meet PLOS ONE’s publication criteria as it currently stands. Therefore, we invite you to submit a revised version of the manuscript that addresses the points raised during the review process. All five reviews were  broadly positive and some minor changes and clarifications are requested. Please note that two reviews were uploaded as attachments. Aditionally, please include a STROBE checklist including page numbers in your revision: https://www.strobe-statement.org/checklists/

We look forward to receiving your revised manuscript.

Kind regards,

Gareth Hagger-Johnson

Academic Editor

PLOS ONE

Journal Requirements:

Reviewers' comments:

Reviewer's Responses to Questions

**Comments to the Author**

1. Is the manuscript technically sound, and do the data support the conclusions?

Reviewer #1: Yes

Reviewer #2: Yes

Reviewer #3: Partly

Reviewer #4: Yes

Reviewer #5: Partly

2. Has the statistical analysis been performed appropriately and rigorously? 

Reviewer #1: Yes

Reviewer #2: Yes

Reviewer #3: I Don't Know

Reviewer #4: Yes

Reviewer #5: Yes

3. Have the authors made all data underlying the findings in their manuscript fully available?

Reviewer #1: Yes

Reviewer #2: Yes

Reviewer #3: No

Reviewer #4: Yes

Reviewer #5: No

4. Is the manuscript presented in an intelligible fashion and written in standard English?

Reviewer #1: Yes

Reviewer #2: Yes

Reviewer #3: Yes

Reviewer #4: Yes

Reviewer #5: Yes

5. Review Comments to the Author

Reviewer #1: Dear authors,

Your work titled “Courses of depressive symptoms and diabetes incidence among middle-aged and older adults” is insightful research. The findings provide insights into the link between specific courses of depressive symptoms and diabetes incidence. This is an original research that is accessible and engaging to read. Only a few structural issues need to be addressed in the manuscript to better align with the journal's guidelines.

The section named “Methods” should be changed to “Materials and Methods” as required by the journal. In the “Data Source” subsection, the inclusion criteria “c” could be clarified by explaining why waves 1 through 4 were included and why waves 5 through 9 were excluded, since self-reported diabetes status was measured from waves 4 through 9. The “Statistical analyses” subsection would benefit from more detail, as some explanations are currently only provided in the table notes.

Regarding the tables, the journal requires that each table be placed directly after the paragraph in which it is first cited. Please also check that the tables are correctly formatted according to the journal’s guidelines.

The citations within the text need correction, as reference numbers should be formatted in square brackets.

Otherwise, the manuscript is well-presented.

Reviewer #2: Excellent written manuscript with a clear justified background, well presented results and discussion. The potential limitation of self-reported diabetes which might have led to missing undiagnosed diabetes has been well studied a study limitation

Reviewer #3: Thank you for the Authors about this interesting manuscript. I wish you all the best with your important work.

As an attachment there are the reviewers comments which hopefully can help to improve the manuscript at least to some extend.

Reviewer #4: First of all, I would like to thank you for the opportunity to review this valuable manuscript. The study addresses an important and timely topic, exploring the association between trajectories of depressive symptoms and diabetes incidence, and provides meaningful insights into this area of research.

1. Strengths and Contributions

The study utilizes a large sample from the English Longitudinal Study of Ageing (ELSA) and a 10-year follow-up period, which enhances the reliability of its findings.

Its originality lies in analyzing different trajectories of depressive symptoms over time (e.g., chronic, remitted, variable), expanding upon existing literature that often focuses solely on cross-sectional or single-time-point assessments of depression.

The inclusion of sensitivity analyses (using different CES-D cut-offs) is a significant strength, demonstrating the stability of results under varying criteria for depressive symptoms.

The study highlights practical implications by suggesting that different trajectories of depressive symptoms may necessitate tailored preventive interventions for diabetes.

2. Minor Observations

While education level was included as a control variable in the analyses, it would be valuable to elaborate further on how it influenced the associations between depressive symptoms and diabetes. For instance, individuals with chronic depressive symptoms had markedly lower educational levels, which could play a critical role in their health outcomes. It is worth noting that lower education levels may impact the risk of both depression and diabetes through mediating factors such as lifestyle, diet, and access to healthcare.

The link provided for reference 2 is incorrect. It does not work when pasted into a browser and should be updated or verified.

3. Suggestion for Additional Analysis

Conducting an interaction analysis between education level and depressive symptom trajectories could provide deeper insights. It may reveal whether the impact of depression on diabetes risk differs based on educational attainment, further enriching the study's findings.

Thank you for considering my feedback. This manuscript contributes significantly to the understanding of the nuanced relationship between depressive symptoms and diabetes incidence.

Reviewer #5: Courses of depressive symptoms and diabetes incidence among middle-aged and older adults.

Evaluation & Suggested Improvements:

1) Abstract:

1. Add more context to the background about the courses of depression with time affects the physical health with diabetes among elderly population based on existing studies.

2. Mentioning the study design, the age of the study population and the type of study (for example Community based study) would have a clear idea about the method. While statistical tools are mentioned, it does not mention which Statistical software was used with their version and license.

3. Include more specific figures or effect sizes (in terms of percentage or fold increases) in the results. To be more complete, the Specific courses of depressive symptoms be mentioned with their ICD-10/DSM-5 codes.

4. Strengthen the conclusion by recommending targeted interventions based on the study findings.

Overall, both the title and abstract are clear but could be refined to be more precise, academic, and impactful.

2) Introduction:

1. To improve comprehension from a border viewpoint, emphasize on the operational definitions for depressive symptoms as research-based framework, ideally aligned with the DSM-5 or ICD-10 while incorporating specific terms like "variable course depressive symptoms" and "remitted course depressive symptoms". Similarly Incident diabetes or Type 2 diabetes mellitus has to be fixed throughout the study and defined scientifically with their standard diagnostic procedures established by the WHO guidelines.

3) Methods:

1. What sample strategy was employed while face to face interview? Were the paper-based questionnaires well-structured and validated by experts to preserve the study's validity and reliability over the course of the ten-year follow-up involving middle-aged and older persons.

2. Along with the mental and physical health status, were the other anthropometric measurements including systolic and diastolic blood pressure, BMI and waist and hip size recorded during the follow up session? As these parameters shows different patterns with age which affects the overall health status including incident diabetes among the elders.

3. To uphold scientific standards, define incident diabetes in accordance with the American Diabetes Association's recommendations or the WHO standard procedure, coordinating their codes with the DSM-5 or ICD-10 and the methodology followed during the participant's diagnosis.

4. Where the research ethics approvals and written informed consent from participants renewed at every follow-up session?

5. Mentioning the inclusive and exclusive criteria would bring more clarity into the study methodology.

6. For better understanding the method used for selection process can be presented with a flowchart.

7. On what basis was the study sample size determined? Was any pilot study carried out while diagnosing for the incident diabetes?

8. Under Measures, the 8-item on which the modified version of the Centre for Epidemiologic Studies-Depression Scale (CES-D) is based can be expressed in detail as these behaviours/emotions focuses on core depressive symptoms such as mood, sleep, appetite, and motivation, typically retaining items that best represent the underlying construct.

9. Under covariates, listing out all the potential confronters and potential mediators separately could give a better understanding of the analysis.

10. Why only the socio-demographic variables and not the physical parameters measured (as per the database) at baseline considered and adjusted for confronting factors while running the Cox proportional hazard ratio model?

11. Statistical analysis are carried out systematically and carefully, bearing the main goal of the study. Mention the software or the application used to carry out the statistical analysis with their version and license.

12. Mentioning Cronbach's alpha assesses the internal consistency reliability of the CES-D scale which effectively calculates the scale’s items measure the same underlying construct-depressive symptoms.

4)Results:

Health is defined as a state of complete physical, mental, and social well-being (WHO,1998). Bearing this in mind and considering the broad aims and scopes of the journal – PLOS ONE, it is not fair to consider only one aspect of health.

1.Was the study impeded by the effects of antidepressant medications? If so, how was it handled?

2. Since older age groups are predisposed to neuropsychiatric illnesses, is it possible to draw a line where we can say with conviction that the symptoms of, say, depression are due to diabetes only and not to old age-related other complications?

3. The presence of diabetic complications (BP levels, obesity, anaemia, etc.) or family history does not seem to be associated with depression, why?

4. The results are presented mostly in text form, but there is limited use of graphs or figures to visually represent key findings. For example, presenting a graph comparing different courses of depressive symptoms along the timeline, would make the longitudinal data more engaging and easier to interpret at a glance.

5) Discussion and Conclusion:

1. When the study discusses that “…clinically meaningful courses…” then defining the specific courses of depressive symptoms and incident diabetes based on established diagnostic guidelines (e.g., DSM-5, ICD-10) becomes critical as to ensures consistency in identifying and measuring these conditions across participants.

2. Despite reports of statistical significance (e.g. HR > 50%), the clinical relevance remains underexplored. For instance, while variable depressive symptoms are linked to incident diabetes, the implications for patient care and treatment are unclear. A deeper discussion on the real-world impact is needed.

3. Indeed, it was worth noting to performed sensitivity analyses, as deviating from any standard protocol needs clear justification. Then the same applies when confronting with the potential physical parameters including the anti-depressive drug effect or the family history or the other complications associated with the age of the participants. Justify?

3. The link between depression symptoms and diabetes may be driven by continuous exposure to underlying physiological and/or behavioral mechanisms.” – agreed, but previous research has shown a reciprocal association between diabetes and depression. Justify?

4. Instead of only quoting the references and naming the mechanisms behind the association between diabetes and the trajectories of depressive symptoms, if the association is demonstrated with possible causes such as vascular alterations or hormonal variations in older persons with diabetes would be much recommended in the discussion section as it could strengthen the study's interpretation and depth.

5. Future studies is recommended with directions that could be more specific, focusing on key gaps in the study.

Important Note: Make sure the data is available while checking through the reference 19 for quick access.

6. PLOS authors have the option to publish the peer review history of their article (what does this mean? ). If published, this will include your full peer review and any attached files.

**Do you want your identity to be public for this peer review?** For information about this choice, including consent withdrawal, please see our Privacy Policy .

Reviewer #1: **Yes: ** Diego Andrade

Reviewer #2: **Yes: ** Godfrey Mutashambara Rwegerera

Reviewer #3: No

Reviewer #4: **Yes: ** Andrzej Śliwerski

Reviewer #5: No

---

## [Author Response · Author response to Decision Letter 1]

23 Jan 2025

Thank you for your invitation to revise and resubmit PONE-D-24-44740: Courses of depressive symptoms and diabetes incidence among middle-aged and older adults.

We also extend our sincere thanks to the reviewers. The reviewers’ positive and constructive feedback about the manuscript is greatly appreciated. Incorporating the thoughtful comments made by the reviewers has further strengthened the manuscript. We are encouraged by their enthusiasm for the paper. We have carefully considered and addressed each reviewer comment. Responses to each reviewer comment can be found below.

Thank you again and best wishes,

Rachel Burns on behalf of the authors

Editor comment

1. Additionally, please include a STROBE checklist including page numbers in your revision

Response: We apologize for this oversight. We have included a STROBE checklist.

Journal Requirements

Response: Thank you. We have reviewed the requirements and changed the manuscript accordingly by moving the tables into the text, updating the reference style, and correcting heading levels.

Response: We have consulted with PLOS ONE’s Peer Review Operations Specialist (Jacob Weller) and our data availability statement has been updated accordingly.

Response: The ethics statement is now included in the manuscript.

Response: We reviewed the reference list and none of the cited papers have been retracted.

Reviewer #1 Comments

1. Your work titled “Courses of depressive symptoms and diabetes incidence among middle-aged and older adults” is insightful research. The findings provide insights into the link between specific courses of depressive symptoms and diabetes incidence. This is an original research that is accessible and engaging to read. Only a few structural issues need to be addressed in the manuscript to better align with the journal's guidelines.

The section named “Methods” should be changed to “Materials and Methods” as required by the journal.

Response: Thank you for your enthusiasm. We have changed “Methods” to “Materials and Methods”.

2. In the “Data Source” subsection, the inclusion criteria “c” could be clarified by explaining why waves 1 through 4 were included and why waves 5 through 9 were excluded, since self-reported diabetes status was measured from waves 4 through 9.

Response: Course of depressive symptoms (as measured during waves 1-4) was the exposure variable. Participants were required to complete the measure of depressive symptoms at each wave during the exposure period to be included in the study. We now clarify in the text, “…provided complete data on measures of depressive symptoms during the waves 1 through 4 (i.e., the exposure period)…” Kindly note that Cox proportional hazards models can account for attrition during follow-up (i.e., waves 5-9), so complete data on diabetes status during waves 5-9 were not required.

3. The “Statistical analyses” subsection would benefit from more detail, as some explanations are currently only provided in the table notes.

Response: We now include the following information, which was previously in table notes, in the body of manuscript, “Sex, ethnicity, marital status, education, age, and the percentage of participant who developed diabetes during follow-up were compared across courses of depressive symptoms via chi-square tests and one-way ANOVAs for categorical and continuous variables, respectively” and “An unadjusted model and a model adjusting for covariates (i.e., sex, ethnicity, marital status, education, age)…”

4. Regarding the tables, the journal requires that each table be placed directly after the paragraph in which it is first cited. Please also check that the tables are correctly formatted according to the journal’s guidelines.

Response: Thank you. Changes made.

5. The citations within the text need correction, as reference numbers should be formatted in square brackets. Otherwise, the manuscript is well-presented.

Response: Thank you. Changes made

Reviewer #2 Comments

Excellent written manuscript with a clear justified background, well presented results and discussion. The potential limitation of self-reported diabetes which might have led to missing undiagnosed diabetes has been well studied a study limitation

Response: Thank you for your positive feedback. We agree that undiagnosed diabetes is a limitation and acknowledge this when discussing limitations, “Diabetes status was assessed via self-reported physician diagnosis. Although good agreement between this measure and fasting plasma glucose has been demonstrated (81.5%), results may not extend to those with undiagnosed diabetes.”

Reviewer #3 Comments to authors

Thank you for the Authors about this interesting manuscript. I wish you all the best with your important work. As an attachment there are the reviewers comments which hopefully can help to improve the manuscript at least to some extend.

Guidance for the review comments:

1. Order of review comment: line number (e.g. 25) in the manuscript, possible sitation

("Depression...") from the manuscript and reviewers comments &/ questions

2. Terminology: major = major issue to consider, minor = minor issue to consider

3. Questions after word optional are optional so no comments is an optional answer for

those questions

Comments to the Authors

• Did the authors consider doing sensitivity analysis with higher than four (4) points cut-off for

CES-D? Further how is the cut-off score for the main analysis justified? If the cut-off score

for depressive symptoms is lowered is there a higher risk for other reasons behind the

incident DM than depressive symptoms?

Response: Thank you for this comment. Cut points were those derived from validation work and previous literature. We clarify in the text, “In the main analyses, individuals with total scores of 4 or more were classified as having depressive symptoms and those with scores of 3 or less were classified as not having depressive symptoms based on previous validation work [20]. However, scores of 3 or more are also often used as an indication of depressive symptoms with this measure [20–22]. Thus, sensitivity analyses were conducted in which individuals with scores of 3 or more were classified as having depressive symptoms and those with scores of 2 or less were classified as not having depressive symptoms.”

• In the main model remitted depressive symptoms and variable course depressive symptoms

were associated with incident diabetes after adjusting for covariates. Yet in the sensitivity

analysis the adjusted model indicated that incident depressive symptoms without remission,

variable course depressive symptoms and chronic depressive symptoms were associated

with incident diabetes. The discrepancy between the results is not fully discussed in the

manuscript.

Response: Thank you. We now highlight this in the Discussion, “Third, the pattern of results differed when different CES-D cut-offs were used. In particular, remitted depressive symptoms were not associated with incident diabetes in sensitivity analyses in which a lower CES-D cut-off was used.”

• At least in the discussion section it should probably be mentioned that variable course

depressive symptoms might be related not only for depression but generally to a more

severe disease called bipolar depression - which may have different impact to the DM risk

(reference).

Response: We agree that this could be a possibility, but bipolar disorder is characterized by periods of depression as well as periods of mania or hypomania. Given that mania and hypomania were not measured in this study, we feel that it would be premature to speculate about the links between the variable course symptoms and bipolar disorder in the discussion section.

Ethics statement

• Minor: "Carleton University Research Ethics Board-B also approved these analyses (Project #

117637)." Word "also": Who else approved the analyses? Further does the approval of

analyses mean approval beforehand (protocol) or afterwards (after conducting the analysis

with the original data and having the same results)? "Participants provided informed

consent verbally to participate in the interview portion of ELSA..." Why informed consent

was not taken in writing?

Response: We have clarified the ethics statement. It know reads, “The National Health Service (NHS) Research Ethics Committees approved the ELSA protocol. Carleton University Research Ethics Board-B also approved these analyses before they were initiated (Project # 117637). All participants provided informed consent verbally to participate in the interview portion of ELSA and in writing for other aspects of the study, such as blood draws and administrative linkages. Verbal consent was witnessed and documented by trained interviewers and fieldworkers.” Kindly note that we used secondary data from ELSA, so we were not involved in the decision to gather verbal, rather than written, informed consent. However, the ethics board of the UK’s National Health Service (NHS) approved this protocol.

• Optional: How was the identity of each participant confirmed (previously known, passport

inspection etc.) and what was the education level and speciality (health care?) of the trained

interviewers and fieldworkers? Even further how experienced the interviewers and

fieldworkers what comes to same kind of tasks before. What kind of training was provided

specifically for the task, interview? How many interviewers and fieldworkers the was? Was

there need to replace any workers during the study and if so did the new workers get

excatly the same education? Was the work monitored during the study and if so what kind

of things or issues the was for discussion? How long is the source data preserved?

Response: In the opening of their review letter, the reviewer indicated that “Questions after word optional are optional so no comments is an optional answer for those questions.” In an effort to keep the manuscript to a reasonable length, we opted not to address these issues in the manuscript.

Data Availability

• Minor: What is the reason for data availability restriction? Hypothetically, could the

reviewer have the original data to conduct and scrutinise the analysis for the

reproductability valuation?

Response: This study used secondary data from the English Longitudinal Study of Ageing (ELSA). ELSA data are publicly available from the UK Data Service. Interested parties, including the reviewer, can create a free UK Data Service account and access the data after agreeing to their conditions of use. The precise ELSA datafile that we used for this study is cited in the manuscript. We consulted PLOS ONE’s Peer Review Operations Specialist (Jacob Weller) and our data availability statement has been updated accordingly.

• Optional: What precise prosedures should be followed to get the data for conducting the

analysis and how long time it would take before everything would be ready for the

reproductability valuation?

Response: Interested parties can create a free UK Data Service account and access the data after agreeing to their conditions of use.

Introduction

• 30, minor: "Approximately 10% of adults worldwide have diabetes (1). Diabetes is more

common among middle-aged and older adults relative to other age groups." Consider word

"younger" instead the word "other".

Response: Change made as requested.

• 34, minor: "Elevated depressive symptoms are a risk factor for diabetes;" Consider words

"Depression symptoms" instead "Elevated depressive symptoms"

Response: Change made as requested.

• 34-36, major: Check references 3 and 26, 27 and correct the discrepancy in the manuscript.

Response: We have checked these references (please note that the latter references have been renumbered as references 29 and 30). The references are for meta-analyses on associations between depressive symptoms and diabetes. Only reference 3 was cited in the Introduction because it is the most recent. We now clarify this in the Introduction, “according to the most recent meta-analysis, depression —as measured at one timepoint—is associated with an 18% increased risk of incident type 2 or adult-onset diabetes [3].”

• 45-46, minor: "In particular, the persistence, remittance, and onset of depressive symptoms

have been differentially linked to physical health outcomes." Consider to mention the

following: There might be multiple reasons why the onset of depression relates to the

incidence of DM. Onset can relate to the overall duration of depression as well to the age of

an individual during the onset of depression and to the era (different era, e.g. 1980 vs. 2010,

might indicate different diagnostic tools and treatment possibilities, perhaps different

treatment outcomes etc. as well).

Response: Thank you. We have considered the reviewer’s point. We agree that age of first depressive episode, and treatment options available during that era, could be an interesting avenue for future research. Unfortunately, these data are not available. The measure of depressive symptoms used in this study (modified CES-D) assessed current depressive symptoms. Depressive symptoms were measured in the same years for all participants in this study thereby holding “era” constant. We also agree that duration of depressive symptoms is an important conside

---

## [Editor Report · Decision Letter 1]

12 Mar 2025

Courses of depressive symptoms and diabetes incidence among middle-aged and older adults: A prospective study

PONE-D-24-44740R1

Dear Dr. Burns,

We’re pleased to inform you that your manuscript has been judged scientifically suitable for publication and will be formally accepted for publication once it meets all outstanding technical requirements.

Kind regards,

Gareth Hagger-Johnson

Academic Editor

PLOS ONE
---

## [Editor Report · Acceptance letter]

PONE-D-24-44740R1

PLOS ONE

Dear Dr. Burns,

I'm pleased to inform you that your manuscript has been deemed suitable for publication in PLOS ONE. Congratulations! Your manuscript is now being handed over to our production team.

Kind regards,

on behalf of

Dr. Gareth Hagger-Johnson

Academic Editor

PLOS ONE
